# Facile Synthesis of B/P Co-Doping Multicolor Emissive Carbon Dots Derived from Phenylenediamine Isomers and Their Application in Anticounterfeiting

**DOI:** 10.3390/nano14100813

**Published:** 2024-05-07

**Authors:** Zhiwei Li

**Affiliations:** 1Key Laboratory for Polymeric Composite and Functional Materials of Ministry of Education, School of Chemistry, Sun Yat-sen University, Guangzhou 510275, China; lizhw36@mail2.sysu.edu.cn; 2State Key Laboratory of Optoelectronic Materials and Technologies, Sun Yat-sen University, Guangzhou 510275, China

**Keywords:** phenylenediamine, isomer, multicolor CDs, co-doping, anticounterfeiting

## Abstract

Carbon dots (CDs) possess a considerable number of beneficial features for latent applications in biotargeted drugs, electronic transistors, and encrypted information. The synthesis of fluorescent carbon dots has become a trend in contemporary research, especially in the field of controllable multicolor fluorescent carbon dots. In this study, an elementary one-step hydrothermal method was employed to synthesize the multicolor fluorescent carbon dots by co-doping unique phenylenediamine isomers (o-PD, m-PD, and p-PD) with B and P elements, which under 365 nm UV light exhibited signs of lavender-color, grass-color, and tangerine-color fluorescence, respectively. Further investigations reveal the distinctness in the polymerization, surface-specific functional groups, and graphite N content of the multicolor CDs, which may be the chief factor regarding the different optical behaviors of the multicolor CDs. This new work offers a route for the exploration of multicolor CDs using B/P co-doping and suggests great potential in the field of optical materials, important information encryption, and commercial anticounterfeiting labels.

## 1. Introduction

With the increasing progress and iterations of science and technology, research on carbon materials has attracted increasing attention in recent years. This includes investigations into various carbon-based materials such as graphene, carbon nanotubes, carbon nanofibers, fullerenes, and carbonaceous materials. Since 2000, carbon dots (CDs) materials have gained popularity owing to their exceptional luminescent properties [1,2,3,4,5,6]. In 2004, researchers such as Xu unexpectedly discovered unknown carbon-based fluorescent nanoparticles while separating single-walled carbon nanotubes using arc discharge soot, making the first instance of CD preparation [7]. From then on, researchers began to study this new type of material, especially in the field of luminescence [8,9,10,11]. In 2006, the Sun research team obtained nano-level carbonaceous grains through the laser ablation of carbon targets. After passivation, the prepared carbon particles exhibited significant fluorescence emission, which led to the novel concept of carbon dots being proposed in the scientific community to describe this unusual type of carbon particle [12]. As a general rule, CDs are spherical nanoparticles, typically zero-dimensional carbon nanomaterials with a dimension smaller than 10 nm, consisting of carbon nuclei and specific surface functional groups.

Until now, the various available types of carbon point synthesis can be generally grouped into two categories: top-down and bottom-up [13,14]. Due to the harsh conditions in the top-down preparation of CDs, complex subsequent processing steps are often required to obtain good samples, which is not suitable for large-scale production. Accordingly, the bottom-up synthesis of CDs has lower requirements for reaction equipment and relatively simple and feasible experimental conditions, meaning that it is gradually becoming a commonly used method for preparing carbon dots. The methods for synthesizing CDs from bottom to top include the combustion method, hydrothermal method, thermal decomposition method, microwave-assisted method, and solvothermal method, among others. The traditional hydrothermal method is an economical, convenient, and environmentally friendly preparation method that has a wide range of easily obtainable precursor sources, such as natural products, biomaterials, and small molecule monomers. The carbon dots prepared using the hydrothermal method not only have a relatively simple postprocessing process but also have excellent optical properties. In 2012, researcher Yang and co-workers synthesized carbon dots using a simple one-step hydrothermal technique [15]. However, making the simple precursor using the hydrothermal method can easily lead to random products being obtained in high-temperature and high-pressure experimental reactions, making it difficult to control the specific chemical structure and physical and chemical properties of the obtained CDs. Recent studies have proposed heteroatom doping using elements such as N, S, B, and P to improve the fluorescence properties of CDs compared with those of the nondoped ones or those doped in different ways [16,17,18,19,20,21,22,23]. CDs doped with different heteroatoms will contribute to improving applications in the fields of encryption [24,25,26,27,28,29]. Therefore, we have found an original material for producing carbon dots, namely phenylenediamine. Phenylenediamine has three isomers (o-PD, m-PD, and p-PD), which can generate the sp2 domain and are beneficial for multicolor carbon dots [30,31,32,33,34]. The amino groups not only provide N-doped and hydrophilic functional groups but also offer a heteroatom binding site [14,35,36].

Here, in this detailed work, we explored a new approach to double heteroatoms co-doping for the synthetic route of multicolor CDs to achieve multicolor optical performance with an emission-tunable color. Using phenylenediamine isomers (o-PD, m-PD, and p-PD) as precursors, and commercial boric acid and commercial phosphoric acid as dopants in the coordination process, the specific CDs (L-CDs, G-CDs, and T-CDs) could be acquired using hydrothermal technology at a temperature of 180 °C for 12 h. As displayed in Figure 1, resulting from B and P double heteroatoms doping, the as-prepared CDs of L-CDs, G-CDs, and T-CDs exhibited a lavender color, grass color, and tangerine color under irradiation with 365 nm ultraviolet rays, respectively. Furthermore, inspired by the tunable color phenomenon, we demonstrate that synthetic fluorescent ink containing CDs and CDs/PVA film, through the integration of CDs into the PVA matrix, can be used in anticounterfeiting applications, such as for transmitting encrypted information or anticounterfeiting labels.

## 2. Materials and Methods

### 2.1. Materials

o-phenylenediamine (o-PD), m-phenylenediamine (m-PD), p-phenylenediamine (p-PD), boric acid, phosphoric acid, and polyvinyl alcohol (PVA) were bought from Shanghai Macklin Biochemical Co., Ltd., (Shanghai, China). The dialysis membrane (cutoff, 1000 Da) was purchased from Viskase Union Carbide Corporation American Co., Ltd. (Bedford Park, IL, USA). All chemicals were used in this study as received without further purification. Deionized water was used throughout this study.

### 2.2. Synthesis of Multicolor CDs

In a straightforward process, 2 mL of phosphoric acid and 1.0 g of boric acid were dissolved in 23 mL of deionized water, producing a clear dispersion. A measure of 108.1 mg of o-PD was added into the above solution, which was stirred for 10 min. Then, the resultant precursor solution was transferred into a commercial 50 mL Teflon-lined autoclave, heated at 180 °C for 12 h, and subsequently naturally cooled down to room temperature. The obtained mixture was first collected using centrifugation, filtered with membrane filters with a pore size of 0.22 μm, and then dialyzed using a dialysis membrane (1000 Da) for 24 h. The powders were prepared from the dialyzed solution by freeze-drying. The resulting solids were redispersed in deionized water to obtain solutions of L-CDs. The synthetic procedure for G-CDs and T-CDs was similar to that for L-CDs, except that the precursor o-PD was replaced with m-PD or p-PD.

### 2.3. Preparation of the Encrypted Ink

An aqueous dispersion (50 mg/mL) of CDs can be used as encrypted ink for anticounterfeiting purposes.

### 2.4. CDs/PVA Film Preparation

To obtain the specific CDs/PVA nanocomposite, first, 1 g of PVA was completely dissolved in a 30 mL aqueous suspension of CDs. Then, the above solution underwent constant magnetic stirring for 2 h at 90 °C and was transferred to glass sheets, which were put into an oven at 100 °C for 3 h to evaporate the moisture content. The film was then cut into suitable shapes for future applications.

### 2.5. Characterization

Transmission electron microscopy (TEM) and high-resolution transmission electron microscopy (HR-TEM) images were obtained using a JEM-2100 transmission microscope (JOEL, Tokyo, Japan, 200 KV). The selected mode is low-dose and the exposure time is 0.5 s. X-ray diffraction (XRD) patterns were recorded using a PANalytical EMPYREAN III X’Pert spectrometer. The anode material of XRD is Cu and its radiate wavelength is 1.54 Å. Fourier transform infrared (FT-IR) spectra of the multicolor CDs were measured with a spectrometer (PerkinElmer Frontier, Waltham, MA, USA) with an attenuated total reflection (ATR) mode. X-ray photoelectron spectroscopy (XPS) spectra were obtained using a Thermo-VG Scientific ESCALAB 250 photoelectron spectrometer. The anode material is Al target and the wavelength is 8 Å. The UV–vis absorption spectra were recorded with a Cary 5000 UV–vis spectrophotometer (Agilent Technology (China) Co., Ltd., Beijing, China). The fluorescent spectrum, fluorescence lifetime, and absolute fluorescence quantum yield (FQY) data were investigated using an Edinburgh FLS1000 fluorescence spectrophotometer. The power used for the fluorescence measurements is 450 W. The color coordinates (CIE) were analyzed using Color Calculator software (CIE1931xy.V.1.6.0.2).

## 3. Results and Discussion

The multicolor carbon dots in this study were synthesized using a facile bottom-up approach, which is a simple, accessible, and fast method [37,38,39,40]. The required multicolored carbon dots were synthesized with co-doping phenylenediamine isomers (o-PD, m-PD, and p-PD) using boric acid and phosphoric acid, and the prepared carbon dots were used as an encryption material. The specific situation details are outlined in the following subsection.

### 3.1. Structural Characterization of the Multicolor CDs

The appearance, microstructure, and particle dimensions of the multicolor CDs (L-CDs, G-CDs, and T-CDs) were verified using TEM. The morphology maps in Figure 1a–c clearly indicate that the prepared multicolor CDs are monodispersed nanoparticles without evident agglomeration. The HRTEM graphics in Figure 1d–f show that all three CDs have specific lattice fringes accompanied by ~0.21 nm lattice spacing, which is related to the graphitic carbon (100) diffraction plane, proving that the obtained CDs have a structure analogous to graphite [41]. As shown in Figure 1g–i, the dimension distributions of L-CDs, G-CDs, and T-CDs are normal distributions, with average particle sizes of 3.55 nm, 3.60 nm, and 3.70 nm, respectively. It is evident that the size of CD particles slowly increases along with the luminescence wavelength, which may cause a bathochromic shift in the luminescence wavelength of the multicolor CDs.

The obtained XRD patterns of the specific CDs in Figure 2a show an expansive diffraction peak at approximately 27°, which indicates there are signs of carbonization present [32]. And the peak at this location indicates a random arrangement in the multicolor carbon dots powder structure. The multicolor CDs do not have distinct crystallinity, which may be explained by heteroatom (B or P) doping or sp3 C defects in the carbon structure [18,42,43]. Additionally, the chemical bonds and surface functional groups of CDs were measured by observing the FT-IR spectra shown in Figure 2b. The three CDs showed broad FTIR absorptions from ~2500 to 3650 cm^−1^, assigned to the stretching vibrations of O-H, N-H, and C-H [18,44]. The corresponding peaks at 1610, 1473, and 1403 cm^−1^ are attributed to C=C stretch vibration, N-H stretch vibration, and C-N stretch vibration, respectively [45]. Finally, the characteristic peaks at approximately 1186, 1011, and 864 cm^−1^ resulted from the stretch vibration of P=O, N–P, and B–O bonds, respectively [44,46].

XPS was performed to further explore the functional groups and chemical composition of the multicolor CDs. As shown in Figure 3a–c, the whole XPS spectra of the L-CDs, G-CDs, and T-CDs indicated five characteristic peaks at approximately 533, 401, 285, 192, and 134 eV, for O 1s, N 1s, C 1s, B 1s, and P 2p, respectively. These peaks suggest that the three prepared CDs are composed of C, N, O, B, and P. The contents of C 1s, N 1s, O 1s, B 1s, and P 2p in the multicolor CDs are listed in Table 1. The data show increasing intensities of the C 1s, N 1s, and O 1s peaks with a redshift in PL emission. In Figure 3d–f, the C1s spectra of three dissimilar CDs are deconvoluted into four peaks at approximately 284.4 eV, 285.1 eV, 286.1 eV, and 288.5 eV, which correspond to C-C/C=C, C-N, -O-C=O, and C-O bonds, respectively [47,48,49,50]. The fitting XPS spectra of the N 1s of the multicolor CDs in Figure 3g–i indicate that the N 1s spectra may be mainly decomposed into four peaks at B–N (399.1 eV), pyridine N (400.7 eV), pyrrole N (401.7 eV), and graphitic N (402.4 eV) [51,52]. It is evident that the concentration of graphite N slowly increased in the high-resolution N 1s spectrum from L-CDs to T-CDs. It is also noticed that graphite N doping appeared to decrease the band gap and energy level of excited electrons transition, which could cause an obvious bathochromic shift [53]. The XPS O 1s spectra in multicolor CDs mainly contain three types of oxygen bonds that can be assigned to C=O (531.2 eV), C–O–C (532.5 eV), and C–O/P–O (533.8 eV), as shown, respectively, in Figure 3g–l [30]. Furthermore, in Appendix A, deconvolution of the B 1s spectra of the multicolor CDs expresses B-N, B-O, and B-CO_2_ bonds located at approximately 190.1 eV, 191.4 eV, and 192.8 eV, respectively [54]. The high-resolution P 2p spectra of the multicolor CDs could be deconvoluted into two peaks, exhibiting two peaks at 133.9 eV and 134.9 eV for P-O/P=O and P-N bonds, which are shown in Appendix A [17].

### 3.2. Optical Performance of the Multicolor CDs

The UV–vis absorption and fluorescent spectra of the multicolor CDs (L-CDs, G-CDs, and T-CDs) in an aqueous solution were analyzed to explore their optical properties, which exhibit a noticeable multicolor tunable fluorescence emission. As shown in Figure 4a–c, the UV–vis absorption spectra of the multicolor CDs showed two clusters of UV absorption peaks, one set of peaks at 218, 230, and 235 nm; and another set at 274, 282, and 283 nm, respectively. The stronger absorption peaks of the multicolor CDs are due to π–π* transitions of C=C and C=N bonds in the benzene ring microstructure [53]. And another weaker absorption peak corresponds to the n–π* transition of C=O/C=N [55]. Among L-CDs, G-CDs, and T-CDs, the absorption peak intensity in the ultraviolet area increased, clearly demonstrating that the content of N-containing aromatic rings increased. The diverse absorption bands of the multicolor CDs verified the different surface functional states of the three prepared typical CDs. Under 365 nm UV light, they appeared as lavender-color, grass-color, and tangerine-color fluorescence, respectively, as displayed in Figure 4a–c inset [56]. Figure 4a–c indicates that the definitive excitation wavelengths of the multicolor CDs occur primarily at 344, 458, and 537 nm, and the optimal emitted peaks are located at 414, 515, and 619 nm, respectively. The above information demonstrates that the multicolor CDs have an emission redshift. The fitting 3D mapping of the L-CDs PL excitation and emission wavelengths shown in Figure 4d denotes one excitation center from 300 nm to 400 nm, instigating the two lavender-color fluorescence emission sources (414 nm and 483 nm). Differently, the fitting 3D mapping of G-CDs PL excitation and emission wavelengths has two unclear excitation centers from 300 nm to 500 nm, corresponding to a central emission center (515 nm), as shown in Figure 4e. A 3D mapping of T-CDs is recorded in Figure 4f, which has an obvious excitation center (537 nm) and two unclear mountain peaks of emission centers (619 nm and 675 nm). This excitation–emission 3D mapping for the multicolor CDs is aligned with optimal excitation and emission wavelengths for the multicolor CDs [57]. Meanwhile, the fluorescent quantum yield (FQY) for multicolor CDs was calculated to be 5.2%, 6%, and 4.5%, respectively. The specific optical parameters of the multicolor CDs are shown in Appendix A. Related to the molar absorption coefficients of published data of the other CDs, the data of the prepared multicolor CDs is smaller [58]. The reason is that the purity of the prepared sample has not reached 100%, and there are still some impurities inside. Subsequently, the CIE coordinates for the multicolor CDs are shown in Figure 4g–i, which better explains the connection between optical properties and fluorescence colors. The CIE coordinates were observed to be (0.21, 0.24), (0.24, 0.46), and (0.66, 0.34), respectively. The CIE coordinate maps have a signal of the lavender color, grass color, and tangerine color, consistent with the fluorescence color of CDs observed using a 365 nm UV lamp. Furthermore, Figure 4j–l demonstrates the time-resolved fluorescence decay curves of L-CDs, G-CDs, and T-CDs separately. Through a fitting operation, the mean fluorescent lifetimes of the multicolor CDs were 4.02, 6.54, and 2.85 ns, respectively, which aligned well with previous fluorescence quantum yield data. The different fluorescence lifetimes observed in the multicolor CDs suggest that they have distinct luminescence mechanisms, as indicated by fitting the decay curves using the tri-exponential function. The average lifetimes of the multicolor CDs were calculated using Equation (1):(1)$$\tau_{\mathrm{avg}}=A_{1}e^{\left(-t/\tau_{1}\right)}+A_{2}e^{\left(-t/\tau_{2}\right)}+A_{3}e^{\left(-t/\tau_{3}\right)}$$ From Equation (1), τ_1_ is mainly attributed to electron transition in the carbon core. At the same time, τ_2_ and τ_3_ are attributed to electron transition in the surface state, which suggests multicolor CDs have surface defect states [50,59,60]. The specific data are listed in Table 2.

### 3.3. Multiemission Fluorescent Mechanism of the Multicolor CDs

On account of the acquired morphology and specific optical characterization results, the fluorescent mechanism of the multicolor CDs with alterable PL performance can be put forward. As shown in Figure 5, the multiemission distinguishing features of CDs depend on the synergistic action of the carbon core and surface functional defect state. In this study, heteroatom doping (B and P) was found to be conducive, producing more defective states of energy levels, which enables the capture of photoexcited electrons [61,62]. At the same time, the lattice doping of the heteroatom (B and P) increases the energy level between carbon π and carbon π*, facilitating multiple routes of electron transferring (n → π*) in PL emission. The different doping atom energy levels cause three radiated recombination steps to return to the ground state. Additionally, the synergistic action induced the emission of broader wavelengths via interband crossing, resulting in a redshifted wavelength of PL emission, which is shown as lavender color (414 nm), grass color (515 nm), and tangerine color (619 nm) [53,63,64,65]. Moreover, the rising carbonization degree and surface defects of the CDs play an important critical role in the area of PL emission.

### 3.4. The Multicolor CDs for Encrypted Information and Anticounterfeiting Labels

Given that the prepared CDs (L-CDs, G-CDs, and T-CDs) display excellent optical performance and changeable multicolor fluorescent behavior, these materials have potential applications in the domain of message encryption and anticounterfeiting labels. Figure 6 reveals the information encryption utilization of the obtained three colored CDs (L-CDs, G-CDs, and T-CDs). The three patterns of Sun Yat-sen University were printed on nonfluorescent paper using multicolor CD inks containing L-CDs, G-CDs, and T-CDs, respectively. Furthermore, with the 365 nm UV light turning on, Figure 6a,b clearly indicates that three tunable pattern colors are visible. Once the UV lamp radiation is turned off, these printed patterns cannot be easily identified with the human eye in sunlight. These results are further proof that the fluorescent quantum yield (FQY) of the G-CDs is the best one among the synthesized CDs. In addition, multicolor CDs have been used to fabricate CDs/PVP films. As shown in Figure 6c, corresponding to the colors of the synthesized water-soluble CDs, the three CDs/PVA films also emit different colors under a UV lamp with 365 nm excitation, representing that the CDs/PVA film can be worked as a unique anticounterfeiting label [57]. These optical phenomena of multicolor CDs hold great promise for applications in the field of information encryption and transmission.

## 4. Conclusions

In conclusion, by using phenylenediamine isomers as precursors and adding boric acid and phosphoric acid as doping reagents, we acquired lavender-color (414 nm), grass-color (515 nm), and tangerine-color (619 nm) fluorescence with a simple hydrothermal method. Surprisingly, the skip distance of emission has a large range from purple color to red color. In view of the measurement results, the optical performance of multicolor CDs is ascribed to the cooperative effect of the core state and the surface functional group defect state of B and P doping. The reasonably regulatable fluorescence color of CDs has great commercialization potential in the areas of information encryption and commercial anticounterfeiting labels.

## Data Availability

The data presented in this study are available on request from the corresponding author.

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
