# Peer review of "Facile Synthesis of B/P Co-Doping Multicolor Emissive Carbon Dots Derived from Phenylenediamine Isomers and Their Application in Anticounterfeiting"

_nanomaterials, 2024, doi:10.3390/nano14100813_

Round 1
Reviewer 1 Report (Previous Reviewer 2)
Comments and Suggestions for Authors
I do not understand why the author didn't want to learn how X-ray works.
The X-ray discussion is still wrong.
2dsin(theta)=lambda means d=0.33nm (lambda/sin(27/2 in degree)/2)
5 references are too much at this place (30-32,42,43), just select one.
This peak, usually labelled 002, has nothing to do with in plane distance.
The in-plane distance correspond to 10, and is at around 2theta=42° (see Warren, B. E. (1941). X-ray diffraction in random layer lattices. Physical Review, 59(9), 693.)
This peak 10 is always very weak.
The 002 peak means that some random stacking in the used power are present.
Nothing can be deduced except that the plane spacing correspond of 2 times the van der Walls radius of carbon atoms but it's obvious.
Author Response
Dear Reviewer 1, please see the attachment.

Reviewer 2 Report (New Reviewer)
Comments and Suggestions for Authors
The manuscript of Li Z. on multicolor emissive carbon dots is a remarkable publication, however, many similar publications can be found in the literature , but it worth publishing.
Instead of o-phenylendiamine, please use 1,2 diaminobenzene, for m-diphenylenediamine use 1,3-diaminobenzene etc.
The author suggest utilization of CD-s as anti-counterfering "dye". However, no mention on their photo bleaching if any, some data would be useful in this aspect. Otherwise the paper is well written, the samples were well characterized.
Author Response
Dear Reviewer 2, please see the attachment.

Reviewer 3 Report (New Reviewer)
Comments and Suggestions for Authors
The manuscript by Li describes preparation and characterization of phosphorous and boron containing Carbon dots (CDs) fluorophores from phenylenediamine isomers. The CDs described in the manuscript have potential multicolor applications, thus the manuscript can be published in Nanomaterials after additional experiments described below.
(1) Please provide the brightness (and molar absorption coefficients, which are required for the calculation of the brightness) of the CDs fluorophores in aqueous solution. Please refer to the studies of molar absorption coefficients and brightnesses of Q-dots for details.
(2) Please perform elemental analysis to analyze the composition of CDs and make comparisons with XPS data shown in table 1, if meaningful correlations are found.
(3) Why is the fluorescence quantum yield of G-CDs (16%) ~3 times higher than that of L-CDs (5.2%) and T-CDs (4.5%)? The fluorescence lifetime shows a similar trend. Please provide some plausible explanation for these observations.
(4) Two diagrams shown in the supporting information are equally useful as the other figures shown in the main body. Therefore, they can be integrated in the body of the manuscript.
(5) Please remove yellow text highlight.
Author Response
Dear Reviewer 3, please see the attachment.

Round 2
Reviewer 1 Report (Previous Reviewer 2)
Comments and Suggestions for Authors
The present version is acceptable
Author Response
Dear Reviewer 1,
Thank you, I am excited to say that you are a very responsible reviewer. I am very grateful for your efforts. Thank you for agreeing to accept my manuscript. I will also strive to learn from you.
Best wishes,
Zhiwei Li
Reviewer 3 Report (New Reviewer)
Comments and Suggestions for Authors
The author has responded to all the reviewer’s comments in a professional manner. The article can be published in nanomaterials with minor revision.
It appears that the molar absorption coefficients of CDs are way much smaller than those of many other fluorophores, including Q-dots. To avoid confusion to the readers, please cite the molar absorption coefficients of published data of the other CDs. (e.g., C 2019, 5, 70. https://www.mdpi.com/2311-5629/5/4/70 I.m)
Author Response
Dear Reviewer 3.
Thank you for your help in my manuscript. I have uploaded the revised manuscript. Please see the attachment.
Best,
Zhiwei Li

This manuscript is a resubmission of an earlier submission. The following is a list of the peer review reports and author responses from that submission.
Round 1
Reviewer 1 Report
Comments and Suggestions for Authors
Recommendation: Minor Revision
Report on : Facile Synthesis of B/P Co-doping Multicolor Emissive Carbon Dots Derived from Phenylenediamine Isomers and their An-ti-Counterfeiting application by Zhiwei Li and Jiaxing Lu
The present manuscript involves the synthesis (via the bottom-up method) and the deep characterization of bis phenylenediamines derived fluorescent CDs bearing boron and phosphorous as dopant. Their potential use in the area of information encryption is also described.
The work reports results that are new and appreciable in the field, the characterization is complete and significant, but three main points are fundamental for its publication:
1. The English language is in some sentence really not understandable. It has to be totally reviewed, also from a grammatical point of view, by a native speaker.
2. I find the doping of the phenylenediamine CDs with B and P a nice idea, but very few are the references regarding the synthesis of the non-doped ones, or doped in different ways. I easily found in the literature many recent works about these themes. Then, some references regarding this subject are absolutely necessary to frame the issue and make people understand the added value of the work presented.
3. In order to render the Results and discussion part self-consistent, a paragraph has to be added at the beginning of part three, dealing with the discussion about the nanodots design and realization (and maybe here some references about them can be added).
Comments on the Quality of English Languagenot good, as reported in point 1.
Author Response
Dear reviewer,
Thank you for your valuable feedback and constructive comments on my manuscript. We have carefully considered your suggestions and tried our best to make the changes of the manuscript. The green part that has been revised according to your comments. Revision notes, point-to-point, are given as follows:
Answer 1. The English language is in some sentence really not understandable. It has to be totally reviewed, also from a grammatical point of view, by a native speaker.
Reply: Please forgive me for the inappropriate use of English sentences in my article. We have rechecked the grammar and formatting of the article. In addition, we have used the editing services of the MDPI to make the article fluent in English writing.
Answer 2. I find the doping of the phenylenediamine CDs with B and P a nice idea, but very few are the references regarding the synthesis of the non-doped ones, or doped in different ways. I easily found in the literature many recent works about these themes. Then, some references regarding this subject are absolutely necessary to frame the issue and make people understand the added value of the work presented.
Reply: Thank you for providing references on further improving non-doped CDs, which is a very meaningful and promising suggestion. I have added some representative references to my article. (line 61)
Answer 3. In order to render the Results and discussion part self-consistent, a paragraph has to be added at the beginning of part three, dealing with the discussion about the nanodots design and realization (and maybe here some references about them can be added).
Reply: Thank you for your constructive suggestions, which by adding some paragraphs about nanodots design can help improve the quality and promotion of the article. I have added some paragraphs about nanodots design to my article. (line 136)
Thank you very much for your attention and time. Look forward to hearing from you.
Yours sincerely,
Zhiwei Li
2024.04.04
Reviewer 2 Report
Comments and Suggestions for Authors
This article reports the luminescence of carbon quantum dots doped with nitrogen and boron.
The authors need to complete and check several parts of the manuscript.
1/ part 2.5 Characterization
a/ What is the voltage for TEM experiment?
b/ What is the wavelength of the X-ray setup (anode material)?
c/ What is the power used for the fluorescence measurements?
2/ line 140: how is obtained 0.21nm? Calculated with Bragg law but what is the anode material?
The shape is symmetric so I do not believe that 10 (always really weak) is possible. In the introduction, the term carbon material was well used so, I do not understand why the authors shifted to graphitic.
A graphite is a tri-dimensional (3D) material with usually AB stacking. This is not the case here. If they assign the peak to 100, it means that's 3D, otherwise, it's 10.
known for nearly one century! (see Biscoe, J., and B. E. Warren. "An X‐ray study of carbon black." Journal of Applied Physics 13.6 (1942): 364-371.)
If the anode is Cu, the peak is just plane spacing between some stacked layers and thus corresponds to something around 0.34-0.36nm (to be calculated).
Citing article (ref34) citing another article (ref12) claiming opposite attribution (0.34nm !) is not serious.
Please, check all references and all the claims before resubmitting.
3/ I do not understand the figure 4. Lavender in their explanation is due to 2 colors, confirmed by light emission.
a/Add the ticks in figure 3a, 3b, 3c to know if the scale is linear or logarithmic
b/Explain how is obtained the curve Ex, from the text, I don't know if it's from the maximum or from the whole PL emission.
c/In Figure 4, the term graphitization is used. This use is incompatible with IUPAC. There is a size increase, nothing else. Carbonization if they want but not graphitization.
d/ typo: line 202: prpared, e is missing
Round 2
Reviewer 2 Report
Comments and Suggestions for Authors
The question about the apparatus was not only for the referees, but also for the readers, the technical information should be added to the text!
For carbon material, a voltage below 80kV allows to avoid knock-out damage. For higher voltages, the exposure time should be very short to avoid changes. So also give the exposure time.
Unfortunately the authors didn't read my comments about X-rays in detail.
Adding lots of references does not change the physics. If something is wrong, it remains wrong. I'm not the alleged referee, I am not in any cited papers. I'm only interested in what is right...
If the wavelength is 8A, with 2theta=27°, the Bragg law (2dsin(theta)=lambda) gives d=17A.
The manuscript is still inconsistent, without enough technical informations so unpublishable in its present form.
Author Response
Thank you for your concern regarding my manuscript. I am extremely grateful. Please see the attachment.

Round 3
Reviewer 2 Report
Comments and Suggestions for Authors
Do the authors understand X-ray diffraction?
They wrote:
"The obtained XRD patterns of the specific CDs in Figure S1 (a) show an expansive diffraction peak at approximately 27°, which indicating the disordered carbon structure of CDs in the field of carbon dots [30-32, 42, 43]"
This corresponds to the interlayer distance between QCD planes, nothing to do with disorder!
They can comment on height through the Scherrer's formula if they want.
The authors didn't understand my first comment (with a reference paper) nor the references they gave!
And what is the meaning of:
"The power used for the fluorescence measurements is xenon lamp"
A power is in W.
Of course, all my previous comments still hold. Impossible to know which scale (linear or logarithmic) is used in figure 3.
Figure 4 not consistent with figure 3 as in figure 3, two bands seem visible for each system.
Wrong graphitization term in figure 4.
As the paper is not seriously checked, rejection is the only option to give the authors time to really correct their manuscript.
Author Response
Dear reviewer, help me to check it suitable or not. I very grateful for you. Please see the attachment
